# Bioconversion of CO to formate by artificially designed carbon monoxide:formate oxidoreductase in hyperthermophilic archaea

Jae Kyu Lim [1,2✉], Ji-In Yang [1,2], Yun Jae Kim [1], Yeong-Jun Park [1,2] & Yong Hwan Kim [3]

Ferredoxin-dependent metabolic engineering of electron transfer circuits has been developed to enhance redox efficiency in the field of synthetic biology, e.g., for hydrogen production and for reduction of flavoproteins or NAD(P)$^+$. Here, we present the bioconversion of carbon monoxide (CO) gas to formate via a synthetic CO:formate oxidoreductase (CFOR), designed as an enzyme complex for direct electron transfer between non-interacting CO dehydrogenase and formate dehydrogenase using an electron-transferring Fe-S fusion protein. The CFOR-introduced *Thermococcus onnurineus* mutant strains showed CO-dependent formate production in vivo and in vitro. The maximum formate production rate from purified CFOR complex and specific formate productivity from the bioreactor were $2.2 \pm 0.2$ μmol/mg/min and $73.1 \pm 29.0$ mmol/g-cells/h, respectively. The CO-dependent $CO_2$ reduction/formate production activity of synthetic CFOR was confirmed, indicating that direct electron transfer between two unrelated dehydrogenases was feasible via mediation of the FeS-FeS fusion protein.

[1] Korea Institute of Ocean Science and Technology (KIOST), Busan 49111, Republic of Korea. [2] KIOST School, University of Science and Technology (UST), Daejeon 34113, Republic of Korea. [3] School of Energy and Chemical Engineering, Ulsan National Institute of Science and Technology (UNIST), Ulsan 44919, Republic of Korea. ✉email: j.k.lim@kiost.ac.kr

Electron transfer is central to various essential metabolic pathways. For example, the electron transport chain and Fe-S proteins are essential constituents of the respiratory complex in all life forms on Earth. Fe-S proteins, which are involved in enzyme catalysis, regulation, maintenance of protein structure, and biological electron transfer[1,2], include [2Fe-2S], [3Fe-4S], and [4Fe-4S] types[1,3–5]. Quantum-mechanical electron transfer between Fe-S clusters could explain the maximum distance of 14 Å for physiologically relevant electron tunneling[6]. This distance has been unable to provide artificially until now and has been observed only in Fe-S proteins interacting each other in nature. Unlike other electron carriers, Fe-S proteins are used as a specific electron path, such as electric wire, so they are attractive candidates for direct electron transfer without electron loss. Ferredoxins are small (~11 kDa) soluble electron carriers that bind to proteins that contain electrons held by reduced Fe-S clusters, providing unique opportunities for engineering and synthetic biology applications[7–13]. Artificial fusion of Fe-S proteins, such as ferredoxin and thioredoxin-like proteins, has been performed to improve electron transfer between redox partners[7–10,14–17]. Despite the reliable efficiency of protein fusion for facilitating electron transfer, target enzymes of fusion constructs are restricted to specific redox pairs that naturally interact with each other, such as ferredoxin and ferredoxin-dependent hydrogenase. Thus, a synthetic electron transferring path between two non-interacting redox enzymes has not yet been reported.

In this study, we attempted direct electron transfer between two different oxidoreductases, carbon monoxide dehydrogenase (CODH) and formate dehydrogenase (FDH), as a model system. CODH and FDH catalyze oxidoreduction of $CO/CO_2$ and formate/$CO_2$, respectively[18–24]. Theoretically, the overall reaction of the oxidation of CO to $CO_2$ with the reduction of $CO_2$ to formate is thermodynamically exergonic ($\Delta G'^o = -16.5$ kJ/mol). CO oxidation coupled with $CO_2$ reduction to formate by connecting CODH and FDH is an ideal system for monitoring electron flow between the two redox enzymes. The electric current can be easily read-out as formate, and the overall reaction does not require an additional substrate other than only CO. We chose the carboxydotrophic and formatotrophic euryarchaeota *Thermococcus onnurineus* NA1 as a model organism, which grows at 63 °C−90 °C (optimum 80 °C)[25]. *T. onnurineus* NA1 harbors genes encoding both CODH and FDH (*codhB* and *fdh3A*, respectively) and shows high cell resistance against both CO and formate[26–28]. Notably, the hydrogen-dependent $CO_2$ reductase (HDCR) enzyme complex catalyzes $CO_2$ reduction to formate in *Acetobacterium woodii* using hydrogen directly or coupled CODH-ferredoxin indirectly as an electron donor[29]. However, no natural enzymes have been shown to catalyze direct CO oxidation coupled with formate production.

Here, we constructed synthetic carbon monoxide:formate oxidoreductase (CFOR) for direct electron transfer between CODH and FDH using an electron-transferring Fe-S fusion protein in *T. onnurineus*. The synthetic CFOR complex was purified and assayed to assess electron transfer ability in vitro as well as in vivo.

## Results

### Construction of CO/formate bioconversion mutants via molecular fusion of two Fe-S proteins
The redox proteins CODH and FDH from *T. onnrineus* NA1 were systematically engineered through molecular fusion of electron-transferring Fe-S proteins to construct a single redox complex. The *codh* and *fdh3* gene clusters included the *codhABCD* and *focA-fdh3ABC* operons, respectively[26,30] (Supplementary Fig. 1a and Supplementary Table 3). The *codhABCD* operon is responsible for CO-dependent ATP synthesis[31], but the intracellular function of *fdh3ABC*

operon has not been elucidated up to date. CodhA and Fdh3B are homologous to the FDH-N β subunit, FdnH (PDB 1FDI), in *E. coli*. FdnH has electron-transferring 4[4Fe-4S] clusters, and its amino acid sequence repeats the common motif for [4Fe-4S] cluster binding (CxxCxxCxnCP) or its slight variants[22]. Sequence alignment of FdnH collected from solved structure showed that the [4Fe-4S] cluster binding motifs were identical to Fdh3B and CodhA (Supplementary Fig. 2a). However, Fdh3C showed high similarity with the 2[4Fe-4S] cluster binding motif in ferredoxins (Supplementary Fig. 2b). The results suggest that Fdh3B and CodhA or Fdh3C have an extrinsic domain with 4[4Fe-4S] or 2[4Fe-4S] clusters, respectively. Protein structure and in silico analyses suggest that FDH small subunits (Fdh3B homolog) directly interact with the FDH catalytic subunit (Fdh3A homolog) and another Fe-S protein (Fdh3C homolog)[22,32,33]. Therefore, Fdh3B was predicted to transfer electrons from Fdh3A to Fdh3C by connecting them in the complex. The amino acid sequences of CodhAB were also homologous to the CooFS proteins (41.7% and 50.3% identity, respectively) in *Rhodospirillum rubrum*. CooF mediates electron transfer from the CODH catalytic subunit CooS to hydrogenase and interacts directly with CooS[34,35]; hence, spontaneous enzyme complex formation of CodhA and CodhB is easily predictable. Accordingly, Fdh3C-CodhA and Fdh3B-CodhA were designed and constructed.

Fusion of Fe-S proteins led to the formation of protein complexes associated with CODH and FDH, termed synthetic CFOR. Therefore, the *fdh3B* or *fdh3C* genes were fused directly to the *codhA* gene using Gibson Assembly in two possible arrangements, *fdh3BC:codhA* and *fdh3B:codhA* (Supplementary Fig. 1b, c). Structurally, the N- and C-termini of FdnH are located on the distal-end [4Fe-4S] cluster[22]; thus, the distal-end [4Fe-4S] cluster at each Fe-S protein was expected to be aligned face-to-face in every possible fusion combinations. Predicted models of the synthetic CFORs are presented in Supplementary Fig. 3. Next, the two flexible linkers $(GGGGS)_1$ and $(GGGGS)_2$ were designed with the *fdh3BC:codhA* fusion arrangement. However, $(GGGGS)_3$ insertion was obtained during the homologous recombination of the $(GGGGS)_2$, resulting in three different lengths of linkers, which was confirmed by sequencing analysis. The *fdh3BC:codhA* fusion constructs pFd3CoL1C1118, pFd3CoL2C1119, and pFd3CoL3C1120 carried three different lengths of flexible linkers, i.e., $(GGGGS)_1$, $(GGGGS)_2$, and $(GGGGS)_3$, respectively (Supplementary Table 1). Notably, the shortest linker, $(GGGGS)_1$, showed the highest formate productivity (Fig. 1b, c) and was selected for *fdh3B:codhA* fusion.

A fosmid vector was used to facilitate cloning for chromosomal insertion of the synthetic CFOR. Insertion of the expression construct was targeted to a region of the chromosome between convergent genes TON_1126 and TON_1127, as previously described[36]. A 9-kbp DNA fragment containing the *fdh3* and *codh* region and the $P_{0157}$ promotor-HMG cassette was inserted into the chromosome of *T. onnurineus* D02 by transformation of pFd3CoL1C1118, pFd3CoL2C1119, and pFd3CoL3C1120, generating the mutant strains BCF01, BCF02, and BCF03, respectively (Supplementary Fig. 1b). Strain BCF13 was then constructed by transformation of the pFd3NStrepCoL1C1149 fosmid, which contains the *fdh3B:codhA* fusion with $(GGGGS)_1$, into *T. onnurineus* D04 strain with additional deletion of the *fdh3* whole gene cluster (Supplementary Figs. 1c and 4a). An affinity-purification Strep-tag was also inserted within the operon at the N-terminus of *fdh3A* to allow easy purification of the synthetic CFOR enzyme at the strain BCF13 (Supplementary Fig. 1c). Strain D05 was also constructed as a negative control for BCF13, which has no fusion linker between *fdh3B* and *codhA* (Supplementary Fig. 4b). To purify the CodhAB sub-complex, strep-tag was fused to N-terminus of CodhB, CODH catalytic subunit, and

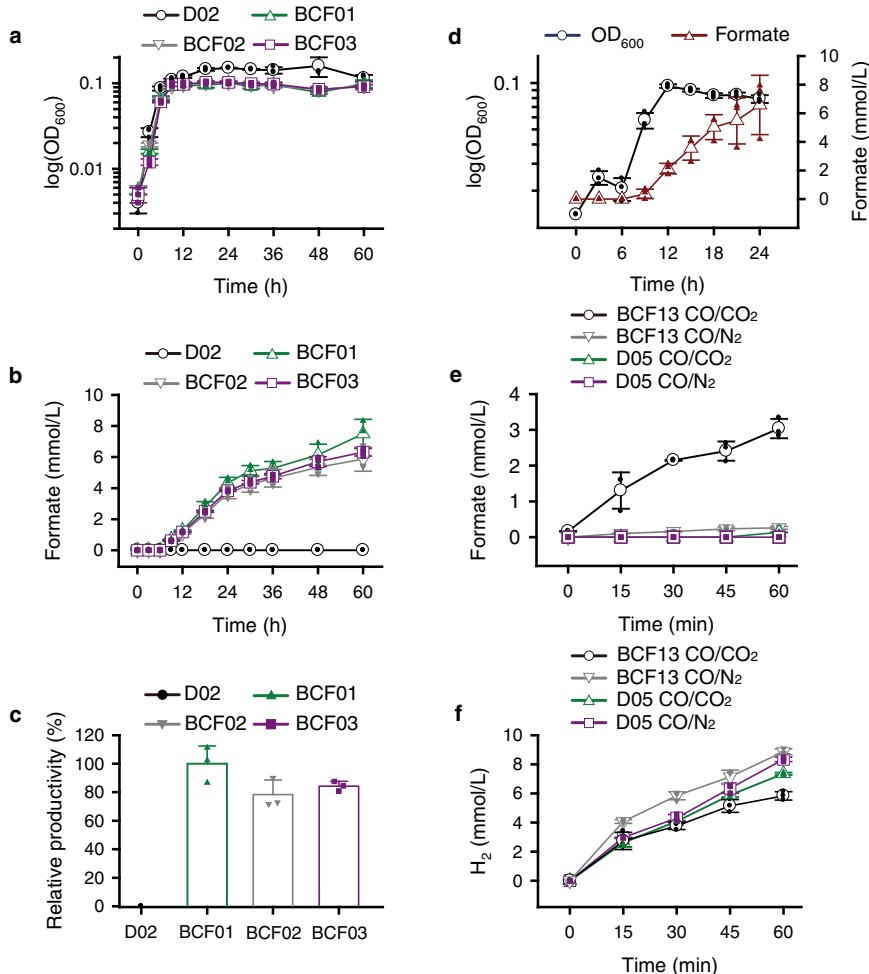

**Fig. 1 CO-dependent formate production. a, b** Cells were incubated at 80 °C with CO gas in serum vial. Cell growth (**a**) and formate production (**b**) of the *T. onnurineus* D02 (parental strain), BCF01 ([GGGGS]$_1$), BCF02 ([GGGGS]$_2$), and BCF03 ([GGGGS]$_3$). **c** Relative formate productivity at 24 h has compared between BCF01 set to 100% and the other strains. **d** Cell growth and formate production from *T. onnurineus* BCF13. **e, f** Formate (**e**) and H$_2$ (**f**) production from cell suspensions of *T. onnurineus* BCF13 (with linker) and *T. onnurineus* D05 (without linker); cells (OD$_{600}$ = 0.5) were incubated with CO/CO$_2$ (50:50 v/v) mix gas or CO/N$_2$ (50:50 v/v) mix gas in modified PBS buffer at 80 °C. The data were presented as the average ± standard deviation (SD) (open symbol), and all the individual data points were shown (closed symbol). Error bars represent ± SD (*n* = 3).

then transformed into *T. onnurineus* D06 strain to construct *T. onnurineus* D07 (Supplementary Table 1).

In summary, strains BCF01, BCF02, and BCF03 harbor Fdh3BC-CodhA fusions of different lengths of (GGGGS)$_1$, (GGGGS)$_2$, and (GGGGS)$_3$, respectively. Strain BCF13 harbors the CFOR complex, composed of Fdh3B-CodhA fusion of (GGGGS)$_1$ linker. The genotype of strain D05, which was constructed to confirm the linker effect, is identical to that of strain BCF13, except that there is no linker between Fdh3B and CodhA. This strain was also used for the purification of the Fdh3AB sub-complex. Strain D07 was used for the purification of the CodhAB sub-complex.

**Determination of CO-dependent cell growth and formate production.** Growth and metabolic profiles of the mutant strains were then compared. Initially, cell growth, formate production, and pH change were determined from *T. onnurineus* BCF01, which contained the fusion of *fdh3C:codhA* with the shortest length of (GGGGS)$_1$ linker. The growth rate of this strain was similar to that of the parental strain, whereas maximum cell growth was slightly reduced (Supplementary Fig. 5a). Cumulative formate production was detected during cell growth with a final

concentration of 3.2 ± 0.2 mmol/L after 60 h of incubation for the BCF01 strain, whereas formate levels were below the detection limit for the parental strain (Supplementary Fig. 5b). Based on the overall reaction, equivalent amounts of formate and H$^+$ are produced, thereby decreasing the pH during formate production. Indeed, the pH of the cell supernatant was significantly decreased from pH 6.9 to 5.0 only in the BCF01 strain (Supplementary Fig. 5c). This may affect sustainable formate production and cell growth, which are optimal at pH 6.5 under CO supplementation conditions[28]. Thus, 0.1 M bis-Tris propane (pH 6.5) buffer was added to the cell growth medium to prevent the abrupt pH change due to bioconversion of CO to formate, thus stabilizing the pH without growth inhibition (Supplementary Fig. 6).

Next, we compared the formate production ability of the remaining mutants. Cell growth was identical, and formate production was detected (Fig. 1). BCF01 showed the highest formate production (7.5 mmol/L) after 60 h of incubation (Fig. 1b). The relative formate productivities of BCF02 and BCF03 at the final time point were determined to be 78% and 84%, respectively, compared with BCF01 (Fig. 1c). The (GGGGS)$_1$ linker showed the highest formate productivity; however, the effect on the electron transfer efficiency was insignificant. BCF13, carrying the *fdh3B:codhA* fusion with the

$(GGGGS)_1$ linker, showed $6.6 \pm 2.1$ mmol/L formate production under CO-supplemented cell growth within 24 h (Fig. 1d), which was higher than that of BCF01 ($4.3 \pm 0.4$ mmol/L formate) at the same time. Cell growth was similar to the other strains. Therefore, all subsequent experiments were conducted using the BCF13 strain. Formate production from the cell suspension was also investigated in serum vials using the strain D05 (without fusion linker) and BCF13 (with fusion linker) at an $OD_{600}$ of about 0.5, incubated at 80 °C in the presence of CO gas with 2 bar (gauge pressure) $CO/CO_2$ (50:50 v/v) or $CO/N_2$ (50:50 v/v) mix gas. Activity and stability of the cell suspensions were confirmed by $H_2$ productivity that showed similar values among strains and gas conditions (Fig. 1f). In the BCF13, $3.0 \pm 0.27$ mmol/L formate was produced after 60 min incubation under the $CO/CO_2$ mix gas (Fig. 1e). In contrast, when the headspace was filled with $CO/N_2$, only $0.3 \pm 0.04$ mmol/L formate was produced due to the low $CO_2$ partial pressure (Fig. 1e and Supplementary Fig. 7). Although the CO oxidation reaction provides the equivalent $CO_2$ requirement for the formate production, the additionally supplemented $CO_2$ enhances the $CO_2$ reduction/formate production reaction. In the D05, formate production was detected under $CO/CO_2$ mix gas as a concentration of $0.1 \pm 0.007$ mmol/L at 60 min, which is 30-fold lower than the BCF13 (Fig. 1e). The results indicate that electron transfer between CODH and FDH modules is achievable just by overexpression of the enzymes but extremely enhanced by a flexible fusion of FeS-FeS in the synthetic CFOR complex.

**Purification of the synthetic CFOR complex**. We then purified the CFOR enzyme complex isolated from strain BCF13 grown in a fed-batch bioreactor with CO-supplemented MM1 medium to activate the expression of genes controlled by the strong promoter $P_{0157}$, which induces robust transcription and translation under CO-supplemented growth conditions[37]. The CFOR complex was purified using a strep-tag fused to the N-terminus of the FDH catalytic subunit, Fdh3A, and then analyzed by SDS-PAGE. Fdh3A, CodhB, and Fdh3B-CodhA fusion subunits, were present, with apparent molecular masses of 76, 67, and 42 kDa, indicating that the Fdh3B-CodhA fusion protein spontaneously bound to CodhB and Fdh3A to form the CFOR complex (Fig. 2a and Supplementary Fig. 8). Protein bands consistent with the calculated molecular weights from deduced amino acid sequences were observed for all three subunits (Supplementary Table 3). The calculated size of Fdh3B-CodhA was 43,722 Da. The CFOR complex was further purified using size-exclusion chromatography, and the CFOR complex from gel filtration was eluted as a single major peak with an apparent mass of around 488 kDa (Fig. 2b, c). The three major bands of purified protein were identified by LC-MS/MS analysis by bands cut from the SDS-PAGE gel (Fig. 2a and Supplementary Table 4). The 76 kDa and 67 kDa protein bands were identified as Fdh3A and CodhB, respectively. The 42 kDa protein band was thought to be the Fdh3B-CodhA fusion protein, which was identified as two different proteins, i.e., Fdh3B and CodhA. The additionally inserted formate transporter FocA[38] could enhance the secretion of intracellular formate synthesized by the CFOR complex. CodhC (29,274 Da) and CodhD (7678 Da) are hypothetical proteins predicted to be involved in the maturation of the catalytic subunit CodhB[39,40]. However, FocA, CodhC, and CodhD subunits were not detected in both the SDS-PAGE gel and by LC-MS/MS analyses. Thus, the entire CFOR complex was composed of the CO dehydrogenase catalytic subunit (CodhB), FDH catalytic subunit (Fdh3A), and Fe-S fusion proteins (Fdh3B-CodhA) connecting the two catalytic subunits. The sub-complexes, Fdh3AB and CodhAB, were individually purified using a strep-tag fused to the N-terminus of each catalytic subunit, Fdh3A and

CodhB, respectively. SDS-PAGE showed Fe-S small subunit, Fdh3B and CodhA, with an apparent molecular mass of 18 kDa and 24 kDa, respectively (Fig. 2a). Based on the molecular weight of CFOR and the information from the protein structure of CODH and FDH[22,41], it seems reasonable to conclude that CFOR has an octameric structure of 4CodhB, 2Fdh3A, and 2Fdh3B-CodhA fusion protein, equivalent to a dimer of 2CodhB, 1Fdh3A, and 1Fdh3B-CodhA tetramers. Therefore the molar ratio of the sub-complexes, Fdh3AB and CodhAB in CFOR, was calculated as 2.2 and 3.8, respectively, which is similar to the protein band intensity of those of CFOR subunits (Supplementary Tables 5, 6).

**Catalytic properties of the CFOR enzyme complex**. The activities of CODH and/or FDH from the purified CFOR, CodhAB, and Fdh3AB enzymes were determined individually using the methyl viologen assay method. The specific CO oxidation activity was determined to be the same level in isolated CFOR and CodhAB as $1,548.5 \pm 366.9$ μmol/mg/min and $1,585.6 \pm 21.8$ μmol/mg/min, respectively. In contrast, there was a huge difference in FDH activity between CFOR and Fdh3AB. The specific formate oxidation activity was determined as $17.1 \pm 0.3$ μmol/mg/min and $2,139.7 \pm 54.4$ μmol/mg/min from the CFOR and Fdh3AB, respectively (Table 1). Why the FDH activity of CFOR complex turns to be extremely low is unclear, but complex formation by protein fusion is thought to be one of the causes of it. In the CFOR, the specific activity of FDH was lower than that of CODH; therefore, the reaction of FDH was expected to be a rate limiting factor for the formate production in the overall CFOR reaction. To confirm direct electron transfer by the Fe-S fusion protein, we investigated whether the purified CFOR could catalyze CO oxidation/formate production without additional electron carriers. The enzyme indeed catalyzed formate production from $CO/CO_2$ (50:50 v/v) mix gas with a maximum specific activity of $2.2 \pm 0.2$ μmol/mg/min at 20 min (Fig. 3 and Table 1). However, formate was below the detection limit under conditions with 100% CO or without CFOR enzyme, which result corresponds to that of the cell suspension experiment (Fig. 1d). In the reaction, $CO_2$ requirement is equilibrium with $CO_2$ generation by CO oxidation; therefore, it could be considered that $CO_2$ partial pressure is not related to the formate production reaction. However, $CO_2$ partial pressure is closely related to the initial formate production rate. When the reaction begins with only CO, generated $CO_2$ by CO oxidation at the CODH active site will be diffused in the solution immediately, rather than reduced at the FDH active site, because a specific $CO_2$ channel is absent in the CFOR system. Therefore, an appropriate $CO_2$ partial pressure is needed for the rapid $CO_2$ reduction at the FDH. As a control experiment for the formate production by CFOR, purified sub-complexes Fdh3AB (18.3 ug) and CodhAB (31.7 ug) were mixed corresponding to 50 ug of CFOR according to the calculated molar ratio (Supplementary Table 5) and then demonstrated the formate production assay under the same condition. After 60 min of reaction, only $0.051 \pm 0.05$ mM of formate was detected in the Fdh3AB and CodhAB sub-complex mixed sample, whereas $2.47 \pm 0.51$ mM, 49-fold higher, was detected in the CFOR complex (Fig. 3). A Fdh3-Codh mixture containing 198 pmol of methyl viologen as the electron carrier corresponding to the FdhB-CodhA fusion protein was also tested, and $0.017 \pm 0.015$ mM of formate production was detected in 60 min. The results demonstrated that the simple, flexible fusion of two Fe-S proteins enabled electron transfer between them, leading to the formation of functional enzyme by an assembly of non-interacting redox enzymes.

**Bioconversion of CO to formate in a bioreactor**. The formate production potential of the strain was tested in a bioreactor where

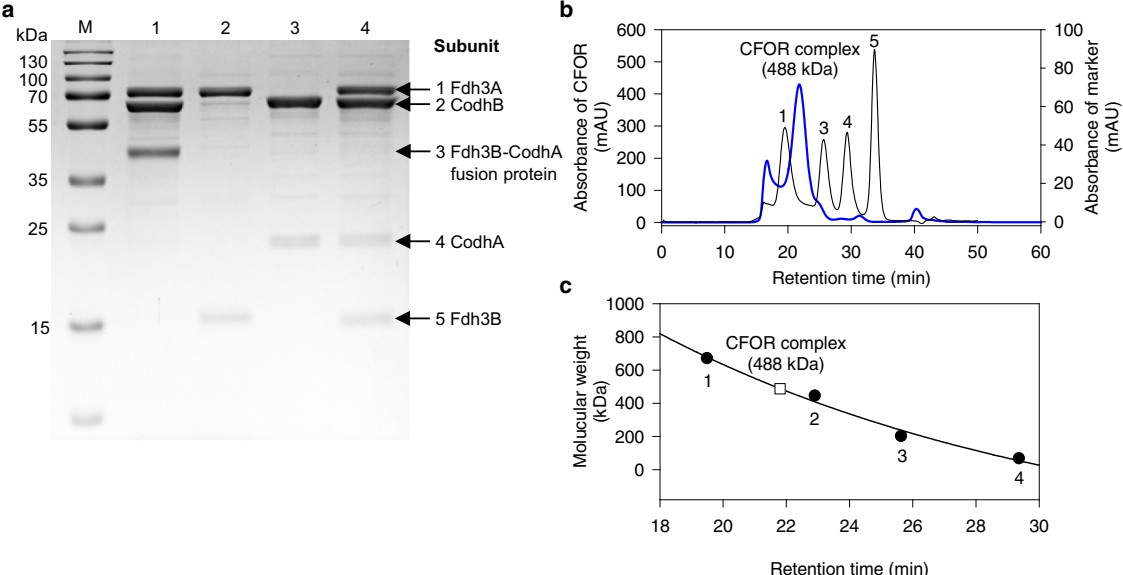

**Fig. 2 Purification and identification of the synthetic CFOR complex from *T. onnurineus* BCF13. a** Strep-tag purified CFOR complex, Fdh3AB, and CodhAB sub-complex were loaded to a 12% acrylamide gel and stained with Coomassie Brilliant Blue after gel running. Subunits are indicated with a black arrow. *M*, molecular mass standards; *lane 1*, CFOR (6 ug); *lane 2*, Fdh3AB (2.2 ug); *lane 3*, CodhAB (3.8 ug); *lane 4*, Fdh3AB (2.2 ug) + CodhAB (3.8 ug). **b** The affinity column purified CFOR complex (blue line), and standard marker (black line) were separated on a Superdex 200 10/300 GL column. A number on a standard peak indicates: 1. Thiroglobulin, 669 kDa; 2. Apoferritin, 443 kDa; 3. Beta-amylase, 200 kDa; 4. Albumin, 66 kDa; 5. Carbonic anhydrase, 29 kDa. **c** The retention time versus the logarithm of the molecular weight of the CFOR complex (open rectangle) and standard marker (closed circle) was plotted on the regression.

**Table 1 Specific activities of CO oxidation, formate oxidation, and formate production from the purified CFOR, Fdh3AB, and CodhAB.**

| Assay | Specific activity (μmol/mg/min)[a] | | |
|---|---|---|---|
| | CFOR | Fdh3AB | CodhAB |
| CO oxidation | 1,548.5 ± 366.9 | n.d. | 1,585.6 ± 21.8 |
| Formate oxidation | 17.1 ± 0.3 | 2,139.7 ± 54.4 | n.d. |
| Formate production | 2.2 ± 0.2 | n.d. | n.d. |

[a]Values of the standard deviation were calculated by three independent experiments.
n.d.: not determined.

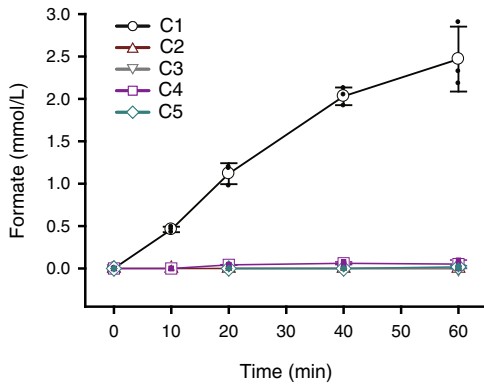

**Fig. 3 In vitro formate production by purified CFOR complex.** The formate production was determined using purified enzyme complex with 2 bar (gauge pressure) of $CO/CO_2$ (50:50 v/v) or 0.5 bar (gauge pressure) of 100% CO in 50 mM sodium phosphate buffer (pH 7.5). Formate production was determined by HPLC. Symbols indicate 50 ug CFOR with 2 bar of $CO/CO_2$ (50:50 v/v) gas (C1); 50 ug CFOR with 0.5 bar of 100% CO gas (C2); mixture of 18.3 ug Fdh3 and 31.7 ug Codh with 2 bar of $CO/CO_2$ (50:50 v/v) gas (C3); mixture of 18.3 ug Fdh3, 31.7 ug Codh, and 198 pmol MV with 2 bar of $CO/CO_2$ (50:50 v/v) gas (C4); absence of enzyme with 2 bar of $CO/CO_2$ (50:50 v/v) gas (C5). The data were presented as the average ± SD (open symbol), and all the individual data points were shown (closed symbol). Error bars represent ± SD ($n = 3$).

100% CO was continuously fed with a flowrate of 0.02–0.122 vvm (CO volume/working volume/min). Table 2 summarizes the-bioreactor parameters for the *T. onnurineus* BCF13 strain. Formate production was detected at a concentration of 56.4 ± 6.4 mmol/L after fermentation for 6 h (Fig. 4b). The formate production rate and maximum specific formate productivity were calculated as 13.1 ± 0.9 mmol/L/h and 73.1 ± 29.0 mmol/g-cells/h, respectively. Recently CO-dependent formate production by coupling of CODH, ferredoxin, and HDCR was reported using *A. woodii* and *Thermoanaerobacter kivui* as a whole-cell biocatalyst; the formate production rate was 0.28 mmol/L/h[42]. The specific rates were determined as 1.44 and 1.34 mmol/g/h for *A. woodii Δrnf* and *T. kivui*, respectively[43]. *T. onnurineus* NA1 and its derivatives mutants used in this study are a basic hydrogenogenic carboxydotroph that can grow on CO as an energy source via the CO-dependent respiratory gene cluster *codh-mch-mnh3*[26,28,30]. Thus, the BCF13 strain showed carboxydotrophic properties, such as $H_2$ and $CO_2$ production (Figs. 1f and 4c), and could grow under 100% CO conditions with the maximum specific growth rate ($\mu_{max}$) of 0.621 ± 0.051 h$^{-1}$ (Fig. 4a). This spontaneous $CO_2$ production by the CO-dependent respiration

enhances the formate production in the bioreactor. Therefore, *T. onnurineus* BCF13 could be used as an industrial microorganism for the production of $H_2$ and formate simultaneously from CO.

## Discussion

Fe-S proteins involved in the electron transfer chain have been well characterized, both structurally and functionally. Fe-S proteins in many oxidoreductase complexes are typically associated

**Table 2 Bioreactor parameters of *T. onnurineus* BCF13.**

| Parameter | Value[d] |
|---|---|
| Maximum specific growth rate, $\mu_{max}$ ($h^{-1}$)[a] | 0.621 ± 0.051 |
| Formate production rate (mmol/L/h)[b] | 13.1 ± 0.9 |
| Maximum formate production rate, $r_{max}$ (mmol/L/h)[c] | 20.7 ± 7.6 |
| Maximum specific formate productivity, $q_{max}$ (mmol/g-cells/h)[c] | 73.1 ± 29.0 |

[a]The $\mu_{max}$ was determined using the values of the linear regression slope in time windows of 1 to 4.5 h.
[b]Values were determined by dividing total yield by time difference from 2 to 6 h.
[c]Values were determined using the difference from the previous time point.
[d]Values of the standard deviation were calculated by four independent experiments.

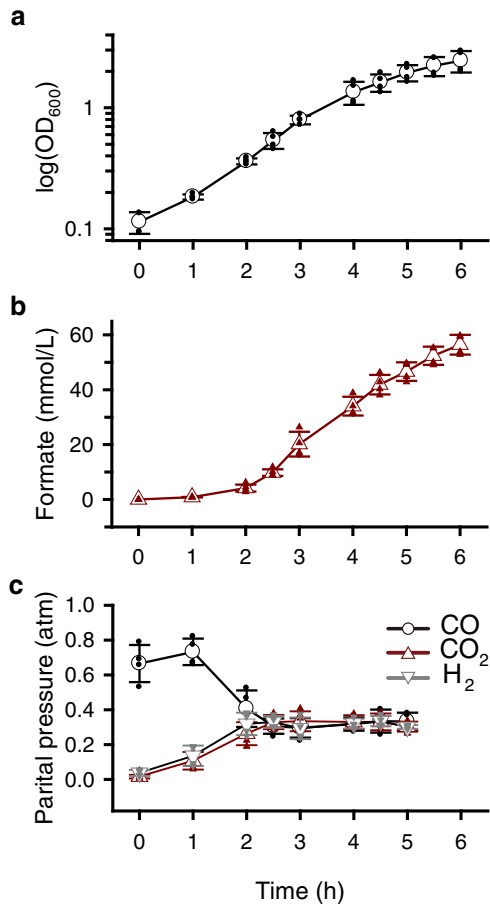

**Fig. 4 CO-dependent fed-batch culture in bioreactor. a** Cell growth was monitored by measuring $OD_{600}$. **b** Formate production was determined by HPLC. **c** Partial pressure of CO, $CO_2$ and $H_2$ in headspace was determined by GC. CO (100%) was supplied with initial flow rate of 0.02 vvm and was raised to 0.122 vvm during the fed-batch culture. The data were presented as the average ± SD (open symbol), and all the individual data points were shown (closed symbol). Error bars represent ± SD ($n = 4$).

with a large subunit that has catalytic redox activity and forms an enzyme complex, such as respiratory complex I, FDH-N and formate hydrogenlyase in *E. coli*[22,32,44]. However, direct electron transfer between CODH and FDH has not yet been found in nature. Thus, if one could construct an electron transfer system between these two oxidoreductases, it could serve as a universal electron transfer system. Accordingly, we generated two different fusion combinations, Fdh3BC-CodhA and Fdh3B-CodhA to, create an electron transfer path, and as a result, $CO_2$ reduction/formate production reaction by electron transport was observed

in all combinations. The small subunits CodhA and Fdh3B specifically interacted with their catalytic large subunits. Therefore, the molecular fusion of Fe-S proteins may spontaneously mediate the formation of a CODH-FDH protein complex (Fig. 2). A significant amount of formate production was only detected from the synthetic CFOR enzyme complex, but not for the individual mixtures of sub-complexes supplemented with or without methyl viologen as an electron carrier (Fig. 3). The results suggest that the electrons from CO oxidation by the CODH are not shared with other electron acceptors and are transferred directly to the FDH, which leads to a concentrated reducing power showing the highest $CO_2$ reduction and formate production rate reported so far.

According to the electron tunneling theory, the maximum distance between the distal end [Fe-S] clusters at each protein must provide a distance of at least 14 Å for electron transfer, indicating that tight-binding and a sophisticated rearrangement of the two proteins are essential, and which was achieved by the flexible linker in the CFOR. Some potential possibilities can be associated with this phenomenon. First, the two sub-complexes have been connected by a flexible linker peptides which allows for motility of the connecting proteins and can move randomly during the reaction[45]. Electrons may transfer if the distal end [Fe-S] clusters of each Fe-S protein coincidently collide by the random motion at an uncertain frequency. Second, the incorporation of Ser residue in the flexible linker can maintain the stability of the linker in aqueous solutions by forming hydrogen bonds with the water molecules[46]. Electron transfer may be induced by these hydrogen bonds mediating rearrangement between the distal [Fe-S] clusters. However, the principles of direct electron transfer by Fe-S fusion protein are unclear, and the validations remain further.

The FDH activity of the CFOR complex significantly decreased than the native Fdh3AB sub-complex (Table 1). It is not clear why the FDH activity is different between the two complexes, but it may result from the conformational change of Fdh3A by protein fusion. Some FDHs from bacteria, including *Desulfovibrio alaskensis*, form heterodimeric $(\alpha\beta)_2$ structures[47]. If Fdh3AB also formed such $(\alpha\beta)_2$ structure, the Fdh3B-CodhAB fusion protein may cause a conformational change in the catalytic moiety, resulting in a lack of FDH activity. Indeed, the CFOR was predicted as a heterodimeric $(Fdh3AB-CodhAB_2)_2$, which can be due to the dimerization of Fdh3AB. To enhance FDH activity, various construction of CFOR using monomeric FDHs and evaluations of enzymatic activity could be necessary.

This work focused on the direct electron transfer between two unrelated redox enzymes by a fusion of Fe-S proteins as an electron path. Using this electron path, we connected non-interacting carbon monoxide dehydrogenase and formate dehydrogenase, constructing a synthetic CFOR as a single functional enzyme complex. The mutant strain BCF13, harbored CFOR encoding genes, showed efficient CO conversion/formate production ability of 73.1 ± 29.0 mmol/g-cells/h. The purified CFOR was also exhibited 2.2 ± 0.2 μmol/mg/min of specific formate productivity from CO. Overall, our results provide some insight into the synthesis of an electron path using the simple fusion of Fe-S proteins, which can be applied to various combinations of redox enzymes for efficient $CO_2$ reduction and production of value-added chemicals.

## Methods

**Strains and cell culture conditions**. *T. onnurineus* NA1 wild-type and mutant strains were routinely cultured in modified medium 1 (MM1)[31,48] containing 4 g/L of yeast extract (BD Bioscience, San Jose, USA) and 4x Holden's trace element/Fe-EDTA solution at 80 °C. All procedures for the cultivation of *T. onnurineus* strains were carried out in an anaerobic chamber (Coy Laboratory Products, Grass Lake, USA). For the cultures in serum bottles, bis-Tris propane (pH 6.5) buffer was additionally supplemented in the medium to a final concentration of 0.1 M, and the

headspaces were filled with 100% CO (MMC) was provided to support the growth of wild-type and mutant strains. The serum bottles were sealed with bromobutyl rubber stoppers and aluminum crimp caps. All procedures were carried out under strictly anaerobic conditions.

For the pH-stat batch culture, *T. onnurineus* strain BCF13 was serially cultured in a 150 ml serum bottle and 7 L bioreactors (Fermentec, Cheongwon, South Korea); the working volumes of which were 80 ml and 5 L of MM1 medium, respectively, at 80 °C. The bioreactors were sparged with pure argon gas (99.999%) through a microsparger. The agitation speed was 500 rpm, and the pH was controlled at 6.2 using 2 M KOH in 3.5% NaCl. The inlet gas of 100% CO was supplied by using a mass flow controller (MKPrecision, Seoul, South Korea) at feeding rates of 100−610 ml/min.

*E. coli* EPI300$^{TM}$-T1$^R$ (Epicentre Biotechnologies, Madison, USA) strain was used for fosmid based molecular cloning purposes. Fosmid containing *E. coli* clones were cultured in LB medium containing 12.5 ug/ml chloramphenicol. General molecular biology manipulations and microbiological experiments were carried out by standard methods[49].

**Cloning and construction of the CFOR expression vector**. The cloning strains, plasmids and fosmids used in this study are listed in Supplementary Table 1. The *fdh3* gene cluster deletion mutant (strain D04) was constructed by the previously used gene disruption system in *T. onnurineus* NA1[36] (Supplementary Fig. 4a). To construct fosmid vector backbone, previously constructed complementary insertion site (Left_arm (TON1128-TON_1127) region-P$_{0157}$ promotor-HMG cassette-Right_arm (TON_1126) region)[36] was amplified by PCR using the primers listed in Supplementary Table 2. The amplicon and fosmid vector pCC1FOS (Epicentre Biotechnologies, Madison, USA) were assembled into a single vector, pNA1-comFosC1096 (Supplementary Table 1), using a SLIC method[50]. PCR products of the *T. onnurineus* NA1 Fdh3 encoding gene region (*focA-fdh3ABC* or *focA-fdh3AB*), Codh encoding gene region (*codhABCD*), and fosmid vector backbone (pNA1comFosC1096 with *Avr*II enzyme digestion) were assembled into a single vector using Gibson Assembly Master Mix (New England Biolabs, Ipswich, USA). The fusion targeting Fe-S protein-encoding genes, *fdh3B* or *fdh3C*, and *codhA* from *T. onnurineus* NA1 were fused using a homologous recombination event with 26 bp complementary PCR primers to generate (GGGGS)$_{1-2}$ flexible linker sequence during the gene assembly reaction by the Gibson Assembly method. The 3′-end of *fdh3B* or *fdh3C* gene lacking its stop codon was fused to the 5′-end of *codhA* gene lacking its start codon mediated by (GGGGS)$_{1-2}$ linker (Supplementary Fig. 1). The sequences of the fusion genes were verified by DNA sequencing. Transformations of *T. onnurineus* strains with the constructed fosmid and the confirmation of transformants were performed by PCR[28]. The primers used to create the fusion genes are listed in Supplementary Table 2.

**Analytical methods**. Cell growth was monitored by measuring optical density at 600 nm (OD$_{600}$) with a spectrophotometer (Eppendorf, Hamburg, Germany). The unit value of OD$_{600}$ corresponded to 0.361 g/L (dry cell weight) as previously determined[37]. The amounts of CO, CO$_2$, and H$_2$ gas were measured using a gas chromatograph (GC; YL Instrument Co., Anyang, South Korea) equipped with a Molsieve 5A column (Supelco, Bellefonte, PA), a Porapak N column (Supelco, Pennsylvania, USA), a thermal conductivity detector, and a flame ionization detector[37]. Formate was measured by high-performance liquid chromatography (HPLC; YL Instrument Co., Anyang, South Korea) using a Shodex RSpak KC-811 column (Showa Denko, Kanagawa, Japan). Ultrapure water containing 0.1% (v/v) phosphoric acid was used as the mobile phase at a flow rate of 1.0 ml/min. All samples were prepared with 1 ml of culture broth and centrifuged to remove cell debris at 4 °C, 13,480 × g for 5 min. The supernatants were purified with a syringe filter and analyzed by HPLC.

**Cell suspension experiment**. To prepare cell suspensions, *T. onnurineus* strain BCF13 was anaerobically cultured in a 7 L fermentor with a working volume of 3 L as described above. At the end of the culture, the cells were harvested by centrifugation at 5523 × g for 30 min at 20 °C and then washed two times with an anaerobic modified PBS (600 mM NaCl, 2.7 mM KCl, 10 mM Na$_2$HPO$_4$, 2 mM KH$_2$PO$_4$, and 2 mM DTT). Finally, cells were resuspended in the same buffer at cell densities of OD$_{600}$ = 0.5. Four milliliters of cell suspensions were transferred to a 20 ml serum vial under a headspace of CO/CO$_2$ (50:50 v/v) mix gas at about 2 bar (gauge pressure), or CO/N$_2$ (50:50 v/v) mix gas at about 2 bar (gauge pressure), respectively. The cell suspensions were incubated at 80 °C, and then the formate concentration and headspace gas composition were determined by HPLC and GC, respectively.

**Purification of enzymes**. To purify CFOR enzyme complex, typically 2 to 4 g (wet weight) of fed-batch cultured *T. onnurineus* strain BCF13 cell pellets were harvested and washed with the modified PBS and then resuspended in buffer W (100 mM Tris–HCl, 150 mM NaCl, pH 8.0). Cells were then disrupted by sonication on ice, and cell debris was removed by centrifugation (15,000 × g for 40 min at 4 °C). Affinity column purification was carried out following the manufacturer's protocols with a Strep-Tactin system (IBA-Lifsciences, Göttingen, Germany). The molecular weight and additional purification of CFOR complex were determined

by analyzing the purified protein on a calibrated Superdex 200 10/300 GL column equilibrated buffer W using fast protein liquid chromatography (Äkta FPLC System, Amersham Biosciences). The column was calibrated by using these standards: thiroglobulin (669 kDa), apoferritin (443 kDa), beta-amylase (200 kDa), albumin (66 kDa) and carbonic anhydrase (29 kDa). Strep-tag purified protein was loaded and eluted at a flow rate of 0.5 ml/min, then the fractions were selectively collected at about 488 kDa of a single peak. The above procedures were carried out under anaerobic conditions. The purified proteins by size exclusion chromatography were used for enzyme assay, and examined via sodium dodecyl sulfate polyacrylamide gel electrophoresis (SDS-PAGE) according to the standard methods. The major bands were identified by LC-MS/MS analysis service (Yonsei Proteome Research Center, Seoul, Korea).

**Enzyme assays**. CODH activity was assayed at 80 °C by a colorimetric assay with methyl viologen (MV) as an electron acceptor ($\varepsilon_{578} = 9.7$ mM/cm at 578 nm)[51], and CO as an electron donor. The assay was conducted with 4.4 ng of purified CFOR complex in 2 ml volume of sodium phosphate buffer (50 mM sodium phosphate, and 2 mM DTT, pH 7.5) containing 5 mM MV in a 4 ml serum-stoppered glass cuvette. 100% CO gas was purged in the headspace at about 2 bar (gauge pressure), and then the reaction was initiated by incubation of the reaction mixture at 80 °C. Formate dehydrogenase activity was assayed using 360 ng of purified CFOR complex under the same conditions except that CO was replaced by 10 mM sodium formate. The formate production assay was carried out in 1 ml volume of 50 mM sodium phosphate buffer (pH 7.5) containing 2 mM DTT and 50 ug of purified CFOR in a 20 ml serum-stoppered vial. CO gas was purged in the headspace at about 2 bar (gauge pressure) of CO/CO$_2$ (50:50 v/v) mix gas, or at about 0.5 bar of 100% CO gas. The reaction was initiated by incubation at 80 °C. Formate in reaction mixture was determined by HPLC.

**Thermodynamic calculation**. The biological standard Gibbs energy value (ΔG′°) was calculated by the Nernst equation using values of the standard Gibbs energy (ΔG°) reported by Amend and Shock[52].

**Statistics and reproducibility**. In all figures, error bars represent the standard deviation of the mean value. To determine product formation from the mutant strains and the CFOR enzyme using HPLC or GC, the number of replicates performed at least three biological replicates ($n = 3$) in every experiment (shown as mean ± s.d.), were measured to reveal a similar level of the product.

**Reporting summary**. Further information on research design is available in the Nature Research Reporting Summary linked to this article.

## Data availability

All source data of the graphs presented in the figures are available in Supplementary Data 1–2 and have been deposited in Figshare (https://doi.org/10.6084/m9.figshare.18851015.v1).

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

## Acknowledgements

This work was supported by the KIOST In-house Program (PEA0022) of the Ministry of Ocean and Fisheries, and the C1 Gas Refinery Program (2015M3D3A1A01064919) through the National Research Foundation of Korea (NRF) funded by the Ministry of Science, ICT & Future Planning. We thank Dr. Kae Kyoung Kwon, Dr. Jung-Hyun Lee, and Dr. Sung Gyun Kang at Marine Biotechnology Research Center, KIOST for supporting this research at the early stage as well as a critical discussion.

## Author contributions

J.K.L. conceptualized the study and study design. The experimental investigation was carried out by J.K.L., J.I.Y., Y.J.K., and Y.J.P. Data analysis was performed by J.K.L. and Y.J.K. Writing of the original draft and editing were carried out by J.K.L. Reviewing was carried out by Y.J.K. and Y.H.K. Supervision was done by J.K.L.

## Competing interests

The authors declare the following competing interests: Patents applications describing the development and applications of CFOR and mutant strains to the KIOST and J.K.L., J.I.Y., Y.J.K., are accepted (no. 10-21928000000, 10-2129279 and 10-2129282) or pending (no. PCT/KR 2018/014807). The remaining authors declare no competing interests.
