## [Peer Review File · Communications Biology]

Reviewers' comments:

Reviewer #1 (Remarks to the Author):

Review for "Bioconversion of CO to formate by artificially designed carbon monoxide:formate oxidoreductase in hyperthermophilic archaea".

This is a very interesting and important study showing a how one can take two thermodynamic distinct chemical reactions, usually shown in microbiology textbooks and via combining their sound thermodynamic potential, literally fuse them together into a real synthetic chimera enzyme, or shall I call it Bizyme? As it performs two completely different reactions at once! The design of this work is very creative, and I also liked the magnetic model, although is completely not backed up by any evidence in this work, nor any suggestions of how to test it occurrence! PLEASE discuss and suggest how you are going to test it.

The flow of the work is not clear, I had to go over and read it again and again to get what was done, therefore, beside a crucial editing step that is prerequisite, I strongly propose the authors to use online software such as "Biorender" to create a "show off" figure which explains the flow of the design and work done here.

The main issue that was not explained well (or shall I say at all) is why is the formate dehydrogenase moiety within the synthetic chimera (please name CFOR as "chimera" or "Bizyme" or any other name which make it clear it is synthetic enzyme) performs so badly, actually 2 magnitude of orders slower! Furthermore, how are you guys going to fix it? For example, via exchanging it with the one from the woodii enzyme? Or somehow, fix the CO₂ canal. Actually, I didn't understand whether its disappearance related to the slow activity?

Here is some specific point to address:

- 1) Why only the short linker worked? If a genuine magnetic field is formed, I would expect that it will work is a range of distances. How this magnetic behavior could be tested in the first place. If you can't measure it, I don't see a place to mention it, regardless of how "sexy" and cool this theory is.
- 2) Page 3 line 11 please cite other works such as came from Poul Erik Jensen and Iftach Yacoby Lab, if I recall correctly, the linker structures and lengths that you are using was mentioned in Yacoby et al 2011 PNAS.
- 3) Page 3 lines 15-17 nice theory! How it can be measure for purified enzyme that you guys have? Please elaborate somewhere....
- 4) Page 5 line 5 "genes" are you refereeing to operons? If yes, please change it into operons and elaborate on the biological function and when does it takes place?
- 5) Page 5, I guess the comparison of each moiety to a different bacterium was done to a solved structure. If that is indeed the case, make it clear! Edit, and maybe add some figure showing the logic flow.
- 6) Page 6 lines 15-17 – This is the explanation of the difference between strain BCF13 and rest. It was better, and you kept using it, hence the difference between it and the other strains should be described more clearly, and not only here.

Wishing you best of luck!

Reviewer #2 (Remarks to the Author):

The manuscript describes a clever and interesting way of coupling between two otherwise not connected enzymes in order to oxidize CO to CO₂ which is reduced in turn to formate. different constructs with connecting peptides were tried and the most efficient peptide for direct electron transfer was a single repeat of GGGGS. The authors have demonstrated a successful coupling. However, I find several claims and conclusions in the study not fully substantiated by evidence and may be less relevant to the experiments that were done in the study. Essentially the major experiment described herein is the combination of different linking peptides fusing between the two enzymes and analysis of the products while optimizing the substrate mixtures that provided to

the enzymatic hybrid enzymatic system. However, in the discussion the authors elude to mechanistic conclusions that should be substantiated by experimental data, otherwise, I would suggest adhering to the enzymatic achievement only and not to the mechanism by which it was achieved as the running title of the manuscript suggests: " Direct electron transfer by a FeS-FeS fusion protein" the electron transfer pathway is unknown and was not investigated at all, also, the suggestion of hyperfine magnetic field generation, should be explained and exemplified mechanistically.

As understood from the manuscript in its current form electron transfer and magnetic field was not studied, so maybe these parts should be tuned down and the enzymatic parameters that were characterized should be highlighted as the generation of a new enzymatic system in a thermophilic organism.

also, please comment on the possibility to conduct the reaction at room temperature if at all as 80-celsius degrees is a very high temperature, and may not be very sustainable for microbial formate production.

Reviewer #3 (Remarks to the Author):

In this manuscript, the authors have synthesized CO:formate oxidoreductase (CFOR) to convert carbon monoxide (CO) gas to formate, the maximum formate production rate and specific formate productivity were $2.2 \pm 0.2 \mu\text{mol/mg/min}$ and $73.1 \pm 29.0 \text{ mmol/g-cells/h}$. Then, the shorter linker ((GGGGS)₁) results in the highest formate productivity, compared with other longer linkers ((GGGGS)₂ and (GGGGS)₃) through experiment. The linkers have been widely used to enhance the performance of the multi-enzyme systems (Critical Reviews in Biotechnology, 2017, 37 (8): 1024-1037), but the mechanism explained by the magnetic property among [Fe-S] clusters look interesting. However, the hypothesis proposed in the discussion section lacks experimental evidence, which is also hard to convince the readers. Although the author constructed a fusion enzyme to convert CO to formate, the reaction mechanism and electron transfer were still unclear. If that is the case, I am not sure that Communication Biology is the most adequate journal. The essential weaknesses include, but are not limited as follow:

1. Author believed that the direct electron transfer between CODH and FDH has not yet been found in nature, so they supposed the proposed mechanism of electron transfer within Fe-S fusion protein by a hyperfine magnetic field. However, the hypothesis lack experiment proof. Thus, the author should design an additional experiment to verify that magnetic fields produce from [Fe-S] clusters and affect the electron transfer, resulting in the higher performance of catalysis.
2. (GGGGS) is a kind of flexible peptide, whose length was non-linear related to the amounts of the amino acid. Thus, the author should get the structure of the three linkers and figure out the relationship between function and structure through experiments or calculations. How about the performance of the linker with rigid peptides?
3. As shown in the results of Fig. 2c, the relative formate productivity between BCF01, BCF02, and BCF03 were a little different, which indicates that the reaction effect of linker on fusion enzyme should be small. This difference may be caused by the rate of diffusion of the substrate and coenzyme, rather than the electron transfer. How does the author exclude this possibility?
4. Author should provide the results about formate and H₂ production from GFOR, in which linkers were (GGGGS)₂ and (GGGGS)₃. If co-express the free enzyme of carbon monoxide dehydrogenase (CODH) and formate dehydrogenase (FDH) without linker in the engineered strain, how about the product performance of formate, compared with the multi-enzyme?

Manuscript COMMSBIO-21-2719-T

Point-by-point response to reviewers

We appreciate the time and effort that the reviewers have dedicated to providing valuable feedback on my manuscript. We are grateful to the reviewers for their insightful comments on my manuscript titled **“Bioconversion of CO to formate by artificially designed carbon monoxide:formate oxidoreductase in hyperthermophilic archaea.”** We have been able to incorporate changes to reflect most of the suggestions provided by the reviewers. We have highlighted the changes within the manuscript. Here is a point-by-point response to the reviewers’ comments and concerns. All page numbers refer to the revised manuscript file.

Note that the page numbers have been changed to match the submitted file merge version, and you can see the original number using Track Changes in the point-by-point response Docx file.

Comments from Reviewer 1

General comment

Review for “Bioconversion of CO to formate by artificially designed carbon monoxide:formate oxidoreductase in hyperthermophilic archaea”.

This is a very interesting and important study showing a how one can take two thermodynamic distinct chemical reactions, usually shown in microbiology textbooks and via combining their sound thermodynamic potential, literally fuse them together into a real synthetic chimera enzyme, or shall I call it Bizyme? As it performs two completely different reactions at once! The design of this work is very creative, and I also liked the magnetic model, although is completely not backed up by any evidence in this work, nor any suggestions of how to test it occurrence! PLEASE discuss and suggest how you are going to test it.

The flow of the work is not clear, I had to go over and read it again and again to get what was done, therefore, beside a crucial editing step that is prerequisite, I strongly propose the authors to use online software such as “Biorender” to create a “show off” figure which explains the flow of the design and work done here.

The main issue that was not explained well (or shall I say at all) is why is the formate dehydrogenase moiety within the synthetic chimera (please name CFOR as “chimera” or “Bizyme” or any other name which make it clear it is synthetic enzyme) performs so badly, actually 2 magnitude of orders slower! Furthermore, how are you guys going to fix it? For example, via exchanging it with the one from the woodii enzyme? Or somehow, fix the CO₂ canal. Actually, I didn't understand whether its disappearance related to the slow activity?

Response: Thank you for the constructive comments on my manuscript. I am also delighted to read that you find that my manuscript makes an important contribution to synthetic enzyme engineering. The name “Bizyme” that you suggested is perfect. I think it represents the novelty of the synthetic CFOR well. However, please understand that it has not been used in the manuscript because I am cautious about suggesting a new term in a single example.

As for the comment that the workflow isn't clear, I think it may come from an ambiguous description of the detailed design of the work, for example, why we used a flexible linker. This research was definitely designed based on the hyperfine magnetic theory. However, the theory was not proved in this study, and reviewers recommended focusing on the enzymatic achievement

rather than the unproved magnetic theory for this paper. Therefore I couldn't describe the detailed work design in the manuscript.

This study began with a CO₂ reduction study. The global warming greenhouse gas CO₂ is thermodynamically stable, making it difficult to convert into reduced molecules. Hence many research efforts have been made to develop an efficient and specific method for biologically converting CO₂. Formate, one of the potential energy storage and chemical feedstock (Hwang et al., 2020; Savage and Zhang, 2016; Yishai et al., 2016), can be produced biologically through the reduction of CO₂ by formate dehydrogenases (FDHs), which reversibly catalyze the oxidation of formate to CO₂ and reduce NAD(P)⁺, quinone, or proton (Jormakka et al., 2002; Lim et al., 2014; Maia et al., 2015; McDowall et al., 2014; Schuchmann and Müller, 2013). However, the reduction of CO₂ to formate by FDHs is tremendous challenging because the reduction potential of these redox partners is similar ($E^{\circ'}_{\text{H}^+/\text{H}_2} = -414 \text{ mV}$) or more positive ($E^{\circ'}_{\text{NAD(P)}^+/\text{NAD(P)H}} = -320 \text{ mV}$) than that of CO₂ ($E^{\circ'}_{\text{CO}_2/\text{Formate}} = -432 \text{ mV}$) (Nelson and Cox, 2005; Thauer et al., 1977). Besides, in the CO₂ reduction by electron carriers except for a direct electron transfer, the reducing power is dispersed to other electron acceptors, resulting in a significant decrease in the CO₂ reduction efficiency. Direct electron transfer from higher energy molecules (such as CO) to CO₂ without electron dispersion is a significant technical challenge to overcome the low-efficiency problem. Utilization of electron transfer Fe-S proteins can be one of the solutions for direct electron transfer. According to the electron tunneling theory, the electron transport conditions between Fe-S proteins are simple: a distance within 14 Å without obstacles between Fe-S clusters. Unfortunately, however, making this simple condition artificially is almost impossible in the nanoscale world. While looking for a way to make electron tunneling condition, I found a report that a hyperfine magnetic field is formed in the reduced Fe-S cluster (Johnson et al., 1969. *Proc. Natl. Acad. Sci. U. S. A.* **63**, 1234–1238), and we hypothesized that the [4Fe-4S] clusters located at both distal ends of the Fe-S fusion protein would combine tightly by physical magnetic interaction, providing the electron tunneling condition (Fig. S7). The flexible linker peptides are used as a kind of string to place Fe-S clusters in a magnetic field. Our magnetic theory is based on proven two facts; first, the reduced Fe-S cluster generates a hyperfine magnetic field. Second, all the molecular magnetic dipoles in the ferromagnetic Fe atoms are pointed in the same direction in response to a magnetic field, resulting in iron being attracted to a magnet. I think that it would have been helpful to understand the strategy and workflow of this study if the magnetic theory could be described detailed in the Introduction. We provide a flowchart below, as you suggested, but have not added it to the manuscript.

($\alpha\beta$)₂ structure, the Fdh3B-CodhAB fusion protein may cause a conformational change in the catalytic moiety, resulting in a lack of FDH activity. Indeed, the CFOR was predicted as a heterodimeric (Fdh3AB-CodhAB)₂, which can be due to the dimerization of Fdh3AB. To enhance FDH activity, various construction of CFOR using monomeric FDHs and evaluations of enzymatic activity could be necessary.”

To improve the Introduction section, some paragraphs have been added and revised as follows:

[page 3, line 32-35]: “This distance has been unable to provide artificially until now and has been observed only in Fe-S proteins interacting each other in nature. Unlike other electron carriers, Fe-S proteins are used as a specific electron path, such as electric wire, so they are attractive candidates for direct electron transfer without electron loss.”

[page 3, line 44-51]: “The [Fe-S] clusters show typical magnetic properties according to the Fe atom's electron spin states, which have been applied to investigate the structure and function of Fe-S proteins using electron magnetic resonance techniques^{2,4}. Johnson *et al.* reported that the iron nuclei in the [Fe-S] cluster showed a magnetic hyperfine interaction with an electron spin S of $1/2$, producing an effective field of about 180 kG in the reduced state¹⁸. Note that the magnetic field strength of 180 kG corresponds to that of a 750 MHz (176 kG) laboratory NMR spectrometer¹⁹. Therefore, we paid attention to the potential possibility that the hyperfine magnetic properties of the [Fe-S] cluster can be utilized to provide the electron tunneling condition.”

Comment 1: *Why only the short linker worked? If a genuine magnetic field is formed, I would expect that it will work is a range of distances. How this magnetic behavior could be tested in the first place. If you can't measure it, I don't see a place to mention it, regardless of how “sexy” and cool this theory is.*

Response: I appreciate your insightful comments and agree with your opinion on the correlation between magnetic fields and distance. We tested three different lengths of fusion linkers (GGGGS)₁, (GGGGS)₂, and (GGGGS)₃, which correspond to the mutant strain BCF01, BCF02, and BCF03, respectively. In the cell growth results (Fig. 2b and 2c), the longer linkers [(GGGGS)₂ and (GGGGS)₃] as well as the shortest linker [(GGGGS)₁] worked and showed varied formate productivity (page 6 lines 101-108). The following paragraph has been added to the Result section

to clarify the difference between mutants (page 6, line 124-page 7, line 129).

“In summary, strains BCF01, BCF02, and BCF03 harbor Fdh3BC-CodhA fusions of different lengths of (GGGGS)₁, (GGGGS)₂, and (GGGGS)₃, respectively. Strain BCF13 harbors the CFOR complex, composed of Fdh3B-CodhA fusion of (GGGGS)₁ linker. The genotype of strain D05 is identical to that of strain BCF13, except that there is no linker between Fdh3B and CodhA to confirm the linker effect. This strain was also used for the purification of the Fdh3AB sub-complex. Strain D07 was used for the purification of the CodhAB sub-complex.”

Regarding the detection of the magnetic behavior, I think that the cryo-electron microscopy (cryo-EM) experiment (this technique is widely used to solve protein structures along with X-ray crystallography) will probably validate the magnetic behavior. If we observe the state of the CFOR enzyme before and after the reaction utilizing cryo-EM, it will be possible to confirm that CODH and FDH are combined by magnetic behavior. I added a more detailed discussion to the revised manuscript (page 12, line 281-page 13, line 292). It read as follows:

“Third, we would carefully suggest that the hyperfine magnetic field may affect this phenomenon. The [4Fe-4S] clusters located at both distal ends of the Fe-S fusion protein may combine tightly by physical magnetic interaction, providing the electron tunneling condition (Supplementary Fig. 7). In that case, the cryo-electron microscopy (cryo-EM) experiment will probably validate the magnetic behavior. According to the proposed magnetic field model, the CFOR is a molecular switch. The hyperfine magnetic field was only detected in the reduced state of [Fe-S] cluster¹⁸. Thus, before the reaction (oxidation state), the two Fe-S proteins are not aligned or tightly bound but are just hanging on together by a flexible linker (Supplementary Fig. 7a). However, after the reaction (reduction state), they would be rearranged and combined tightly under the influence of magnetic strength (Supplementary Fig. 7c). Such conformational changes with oxidation and reduction states are probably observed with cryo-EM images. However, the principles of direct electron transfer by Fe-S fusion protein are unclear, and the validations remain further.”

Comment 2: *Page 3 line 11 please cite other works such as came from Poul Erik Jensen and Iftach Yacoby Lab, if I recall correctly, the linker structures and lengths that you are using was*

mentioned in Yacoby et al 2011 PNAS.

Response: Thank you for the useful reference. It has been included in the revised manuscript (page 3, line 39).

Comment 3: Page 3 lines 15-17 nice theory! How it can be measure for purified enzyme that you guys have? Please elaborate somewhere....

Response: I appreciate that you positively evaluate the magnetic theory. I think such magnetic behavior can be detected utilizing cryo-EM, as I mentioned in comment 1. I added a more detailed discussion to the revised manuscript following your suggestion (page 12, line 281-page 13, line 292).

“Third, we would carefully suggest that the hyperfine magnetic field may affect this phenomenon. The [4Fe-4S] clusters located at both distal ends of the Fe-S fusion protein may combine tightly by physical magnetic interaction, providing the electron tunneling condition (Supplementary Fig. 7). In that case, the cryo-electron microscopy (cryo-EM) experiment will probably validate the magnetic behavior. According to the proposed magnetic field model, the CFOR is a molecular switch. The hyperfine magnetic field was only detected in the reduced state of [Fe-S] cluster¹⁸. Thus, before the reaction (oxidation state), the two Fe-S proteins are not aligned or tightly bound but are just hanging on together by a flexible linker (Supplementary Fig. 7a). However, after the reaction (reduction state), they would be rearranged and combined tightly under the influence of magnetic strength (Supplementary Fig. 7c). Such conformational changes with oxidation and reduction states are probably observed with cryo-EM images. However, the principles of direct electron transfer by Fe-S fusion protein are unclear, and the validations remain further.”

Comment 4: Page 5 line 5 “genes” are you refereeing to operons? If yes, please change it into operons and elaborate on the biological function and when does it takes place?

Response: Thanks for your kind reminders. I revised the sentence as follows (page 5, lines 76-79):

“The *codh* and *fdh3* gene clusters included the *codhABCD* and *focA-fdh3ABC* operons, respectively^{27,31} (Fig. 1a and Supplementary Table 3). The *codhABCD* operon is responsible for CO-dependent ATP synthesis³², but the intracellular

function of *fdh3ABC* operon has not been elucidated up to date.”

Comment 5: Page 5, I guess the comparison of each moiety to a different bacterium was done to a solved structure. If that is indeed the case, make it clear! Edit, and maybe add some figure showing the logic flow.

Response: Yes, I collected the amino acid sequence from the solved structure, and it has been described in the revised manuscript (bold, italic) according to your suggestion. It read as follows (page 5 lines 82-83):

“Sequence alignment of FdnH *collected from solved structure* showed that the [4Fe-4S] cluster binding motifs were identical to Fdh3B and CodhA (Supplementary Fig. 1a).”

I am afraid that I hardly understand the meaning of “showing the logic flow.” I take your point that an explanation of the reason for using the amino acid sequence alignment is required. However, comparing amino acid sequences is a commonly used analytical method to show protein similarity and predict protein function. I think that page 5, line 79-94 (read as below) explains enough about it. Therefore, I ultimately decided not to add logic flow.

“CodhA and Fdh3B are homologous to the FDH-N β subunit, FdnH (PDB 1FDI), in *E. coli*. FdnH has electron-transferring 4[4Fe-4S] clusters, and its amino acid sequence repeats the common motif for [4Fe-4S] cluster binding (CxxCxxCxnCP) or its slight variants²⁴. Sequence alignment of FdnH collected from solved structure showed that the [4Fe-4S] cluster binding motifs were identical to Fdh3B and CodhA (Supplementary Fig. 1a). However, Fdh3C showed high similarity with the 2[4Fe-4S] cluster binding motif in ferredoxins (Supplementary Fig. 1b). The results suggest that Fdh3B and CodhA or Fdh3C have an extrinsic domain with 4[4Fe-4S] or 2[4Fe-4S] clusters, respectively. Protein structure and *in silico* analyses suggest that FDH small subunits (Fdh3B homolog) directly interact with the FDH catalytic subunit (Fdh3A homolog) and another Fe-S protein (Fdh3C homolog)^{24,33,34}. Therefore, Fdh3B was predicted to transfer electrons from Fdh3A to Fdh3C by connecting them in the complex. The amino acid sequences of CodhAB were also homologous to the CooFS proteins (41.7% and 50.3% identity, respectively) in *Rhodospirillum rubrum*. CooF mediates electron transfer from the CODH catalytic subunit CooS to

hydrogenase and interacts directly with CooS^{35,36}; hence, spontaneous enzyme complex formation of CodhA and CodhB is easily predictable. Accordingly, Fdh3C-CodhA and Fdh3B-CodhA were designed and constructed.”

Comment 6: *Page 6 lines 15-17 – This is the explanation of the difference between strain BCF13 and rest. It was better, and you kept using it, hence the difference between it and the other strains should be described more clearly, and not only here.*

Response: Thanks for your kind reminders. The difference between mutants has been added in the revised manuscript as follows (page 6, line 124-page 7, line 129):

“In summary, strains BCF01, BCF02, and BCF03 harbor Fdh3BC-CodhA fusions of different lengths of (GGGGS)₁, (GGGGS)₂, and (GGGGS)₃, respectively. Strain BCF13 harbors the CFOR complex, composed of Fdh3B-CodhA fusion of (GGGGS)₁ linker. The genotype of strain D05 is identical to that of strain BCF13, except that there is no linker between Fdh3B and CodhA to confirm the linker effect. This strain was also used for the purification of the Fdh3AB sub-complex. Strain D07 was used for the purification of the CodhAB sub-complex.”

Comments from Reviewer 2

General comment

The manuscript describes a clever and interesting way of coupling between two otherwise not connected enzymes in order to oxidize CO to CO₂ which is reduced in turn to formate. Different constructs with connecting peptides were tried and the most efficient peptide for direct electron transfer was a single repeat of GGGGS. The authors have demonstrated a successful coupling. However, I find several claims and conclusions in the study not fully substantiated by evidence and may be less relevant to the experiments that were done in the study

Response: I appreciate the time and effort that you have dedicated to providing your valuable feedback on my manuscript. I believe that your valuable and insightful comments led to possible improvements in the current version. We have carefully revised the manuscript according to the Reviewer's comments and concerns and provided point-by-point responses. To improve the Introduction section, some paragraphs have been added and revised as follows:

[page 3, line 32-35]: “This distance has been unable to provide artificially until now and has been observed only in Fe-S proteins interacting each other in nature. Unlike other electron carriers, Fe-S proteins are used as a specific electron path, such as electric wire, so they are attractive candidates for direct electron transfer without electron loss.”

[page 3, line 44-51]: “The [Fe-S] clusters show typical magnetic properties according to the Fe atom's electron spin states, which have been applied to investigate the structure and function of Fe-S proteins using electron magnetic resonance techniques^{2,4}. Johnson *et al.* reported that the iron nuclei in the [Fe-S] cluster showed a magnetic hyperfine interaction with an electron spin S of $1/2$, producing an effective field of about 180 kG in the reduced state¹⁸. Note that the magnetic field strength of 180 kG corresponds to that of a 750 MHz (176 kG) laboratory NMR spectrometer¹⁹. Therefore, we paid attention to the potential possibility that the hyperfine magnetic properties of the [Fe-S] cluster can be utilized to provide the electron tunneling condition.”

Comment 1: *Essentially the major experiment described herein is the combination of different linking peptides fusing between the two enzymes and analysis of the products while optimizing the substrate mixtures that provided to the enzymatic hybrid enzymatic system. However, in the*

discussion the authors elude to mechanistic conclusions that should be substantiated by experimental data, otherwise, I would suggest adhering to the enzymatic achievement only and not to the mechanism by which it was achieved as the running title of the manuscript suggests: "Direct electron transfer by a FeS-FeS fusion protein" the electron transfer pathway is unknown and was not investigated at all, also, the suggestion of hyperfine magnetic field generation, should be explained and exemplified mechanistically. As understood from the manuscript in its current form electron transfer and magnetic field was not studied, so maybe these parts should be tuned down and the enzymatic parameters that were characterized should be highlighted as the generation of a new enzymatic system in a thermophilic organism. As understood from the manuscript in its current form electron transfer and magnetic field was not studied, so maybe these parts should be tuned down and the enzymatic parameters that were characterized should be highlighted as the generation of a new enzymatic system in a thermophilic organism.

Response: Thank you for the constructive comments needed to improve my manuscript. You make a valid point that the paper should focus more explicitly on enzymatic achievement. As you commented, we agree that the suggested theory is not a proven hypothesis. I've rewritten the Discussion section to minimize the magnetic field theory following your suggestion (page 12-14). The magnetic theory is suggested as one of some possibilities, and the figure of the electron transport mechanism (Fig. 6) has been removed from the main manuscript then moved to Supplementary Data (Fig. S7) as a model (page 12, line 271-page 13, line 284). It read as follows:

“According to the electron tunneling theory, the maximum distance between the distal [Fe-S] clusters at each protein must provide at least 14 Å for electron transfer, indicating that tight-binding and a sophisticated rearrangement of the two Fe-S proteins are essential, and which was achieved by the flexible linker in the CFOR. Several potential possibilities can be associated with this phenomenon. First, the two sub-complexes have been connected by a flexible linker which allows for motility of the connecting proteins and can move randomly during the reaction^{17,46}. Electrons may transfer if the distal end [Fe-S] clusters of each Fe-S protein coincidentally collide by the random motion at an uncertain frequency. Second, the incorporation of Ser residue in the flexible linker can maintain the stability of the linker in aqueous solutions by forming hydrogen bonds with the water molecules⁴⁷. Electron transfer may be induced by these hydrogen bonds mediating rearrangement between the distal [Fe-S] clusters. Third, we would carefully suggest that the hyperfine magnetic field may affect this phenomenon. The [4Fe-4S] clusters located at both distal ends of

the Fe-S fusion protein may combine tightly by physical magnetic interaction, providing the electron tunneling condition (Supplementary Fig. 7).”

Comment 2: *Please comment on the possibility to conduct the reaction at room temperature if at all as 80-celsius degrees is a very high temperature, and may not be very sustainable for microbial formate production.*

Response: Thanks for your kind reminders. I take your point that you thought mesophilic bacteria such as *Escherichia coli* for the formate production. I realized that the growth temperature of the model strain, *Thermococcus onnurineus* NA1, was not described in the manuscript. *T. onnurineus* NA1, has been reported as a hyperthermophile that grows in a temperature range between 63°C-90°C, and the optimal growth temperature is 80°C; room temperature (above 25°C) is below the growth temperature (Bae, S. S. *et al.*, 2006. *J. Microbiol. Biotechnol.* **16**, 1826–1831). Therefore, 80°C is a suitable temperature for the sustainable formate production for this strain, and *in vivo* formate production at room temperature cannot be expected. The possibility of conducting the reaction at room temperature is not described in the manuscript, instead, growth temperature has been added in the Introduction as follows (page 4, line 59-60):

“We chose the carboxydophilic and formatotrophic euryarchaeota *Thermococcus onnurineus* NA1 as a model organism, which grows at 63°C-90°C (optimum 80°C)²⁷.”

Comments from Reviewer 3

General comment

In this manuscript, the authors have synthesized CO:formate oxidoreductase (CFOR) to convert carbon monoxide (CO) gas to formate, the maximum formate production rate and specific formate productivity were $2.2 \pm 0.2 \mu\text{mol/mg/min}$ and $73.1 \pm 29.0 \text{ mmol/g-cells/h}$. Then, the shorter linker ((GGGGS)₁) results in the highest formate productivity, compared with other longer linkers ((GGGGS)₂ and (GGGGS)₃) through experiment. The linkers have been widely used to enhance the performance of the multi-enzyme systems (Critical Reviews in Biotechnology, 2017, 37 (8): 1024-1037), but the mechanism explained by the magnetic property among [Fe-S] clusters look interesting. However, the hypothesis proposed in the discussion section lacks experimental evidence, which is also hard to convince the readers. Although the author constructed a fusion enzyme to convert CO to formate, the reaction mechanism and electron transfer were still unclear. If that is the case, I am not sure that Communication

Response: Thank you for the constructive comments on my manuscript. I also appreciate the time and effort that you have dedicated to providing your valuable feedback on my manuscript. I believe that your valuable and insightful comments led to possible improvements in the current version. We have carefully revised the manuscript according to the Reviewer's comments and concerns and provided point-by-point responses.

Thank you for the useful reference. It has been included in the revised manuscript (page 12, line 276). To improve the Introduction section, some paragraphs have been added and revised as follows:

[page 3, line 32-35]: "This distance has been unable to provide artificially until now and has been observed only in Fe-S proteins interacting each other in nature. Unlike other electron carriers, Fe-S proteins are used as a specific electron path, such as electric wire, so they are attractive candidates for direct electron transfer without electron loss."

[page 3, line 44-51]: "The [Fe-S] clusters show typical magnetic properties according to the Fe atom's electron spin states, which have been applied to investigate the structure and function of Fe-S proteins using electron magnetic resonance techniques^{2,4}. Johnson *et al.* reported that the iron nuclei in the [Fe-S]

cluster showed a magnetic hyperfine interaction with an electron spin S of $1/2$, producing an effective field of about 180 kG in the reduced state¹⁸. Note that the magnetic field strength of 180 kG corresponds to that of a 750 MHz (176 kG) laboratory NMR spectrometer¹⁹. Therefore, we paid attention to the potential possibility that the hyperfine magnetic properties of the [Fe-S] cluster can be utilized to provide the electron tunneling condition.”

Below is a point-by-point response to Reviewer 1 comments and concerns.

Comment 1: *Author believed that the direct electron transfer between CODH and FDH has not yet been found in nature, so they supposed the proposed mechanism of electron transfer within Fe-S fusion protein by a hyperfine magnetic field. However, the hypothesis lack experiment proof. Thus, the author should design an additional experiment to verify that magnetic fields produce from [Fe-S] clusters and affect the electron transfer, resulting in the higher performance of catalysis.*

Response: Thank you for the constructive comments needed to improve my manuscript. As you comment, we agree that the suggested theory is not a proven hypothesis. In this study, we focused on the possibility of electron transfer between two different oxidoreductases by fusion of Fe-S proteins, and the validation of the magnetic theory is being planned as further study. Therefore, additional experiments are not currently being considered; instead, I've rewritten the Discussion section to minimize the magnetic field theory and focus on enzymatic achievement. The magnetic theory is suggested as one of some possibilities, and the figure of the electron transport mechanism (Fig. 6) has been removed from the main manuscript then moved to Supplementary Data (Fig. S7) as a model (page 12, line 271–page 13, line 284). It read as follows:

“According to the electron tunneling theory, the maximum distance between the distal [Fe-S] clusters at each protein must provide at least 14 Å for electron transfer, indicating that tight-binding and a sophisticated rearrangement of the two Fe-S proteins are essential, and which was achieved by the flexible linker in the CFOR. Several potential possibilities can be associated with this phenomenon. First, the two sub-complexes have been connected by a flexible linker which allows for motility of the connecting proteins and can move randomly during the reaction^{17,46}. Electrons may transfer if the distal end [Fe-S] clusters of each Fe-S protein coincidentally collide by the random motion at an uncertain frequency. Second, the incorporation of Ser residue in the flexible linker can maintain the stability of the linker in aqueous solutions by forming

hydrogen bonds with the water molecules⁴⁷. Electron transfer may be induced by these hydrogen bonds mediating rearrangement between the distal [Fe-S] clusters. Third, we would carefully suggest that the hyperfine magnetic field may affect this phenomenon. The [4Fe-4S] clusters located at both distal ends of the Fe-S fusion protein may combine tightly by physical magnetic interaction, providing the electron tunneling condition (Supplementary Fig. 7).”

Comment 2: *(GGGGS) is a kind of flexible peptide, whose length was non-linear related to the amounts of the amino acid. Thus, the author should get the structure of the three linkers and figure out the relationship between function and structure through experiments or calculations. How about the performance of the linker with rigid peptides?*

Response: Thank you for the constructive comments on my manuscript. As you commented, we agree that crystal structure of the linker region is very important for understanding the precise mechanism of direct electron transfer as well as the function of the flexible linker in the synthetic CFOR. Till now, we focused on the synthesis of a novel chimeric enzyme by fusing two different Fe-S proteins, and solving protein structure seemed likely out of focus in this study. The crystal structure study using cryo-EM is being planned as further study.

Regarding the performance of the rigid linker, unfortunately, we have no idea about the performance of the rigid linker because it's not been tested till now. The electron tunneling theory says that a maximum distance of 14 Å is essential for the direct electron transfer (Page et al., 1999. *Nature* **402**, 47–52). Therefore, rigid linkers were excluded from the candidates of fusion linkers from the beginning because they definitely have the potential to make a space longer than 14 Å. This issue is also seemed likely out of focus, and we have no plan for an additional experiment about it.

Comment 3: *As shown in the results of Fig. 2c, the relative formate productivity between BCF01, BCF02, and BCF03 were a little different, which indicates that the reaction effect of linker on fusion enzyme should be small. This difference may be caused by the rate of diffusion of the substrate and coenzyme, rather than the electron transfer. How does the author exclude this possibility?*

Response: Thanks for your kind reminders. I get your point very well. However, thinking carefully about this issue, the only genetic difference between mutant strains is the length of the linker peptide. Therefore, it is thought that the physiological change will be insignificant. The experiment was demonstrated at a long time window with the triplicate, and there is seemed to be

no clue whether the rate diffusion of substrate and coenzyme affected the rate of formate production. Therefore, the difference in formate production between mutant strains is thought to be the effect of linker length.

Comment 4: *Author should provide the results about formate and H₂ production from CFOR, in which linkers were (GGGGS)₂ and (GGGGS)₃. If co-express the free enzyme of carbon monoxide dehydrogenase (CODH) and formate dehydrogenase (FDH) without linker in the engineered strain, how about the product performance of formate, compared with the multi-enzyme?*

Response: Thank you for the constructive comments. We agree that the enzymatic assessment of different linker length of CFOR will be interesting research topic. However, the detailed validation of enzymatic linker length is a characterization of the linker itself rather than making a novel electron path, which seems likely a slight step aside from the major concept of this study. Therefore, I would like to carry the validation of linker length on the further study, which will focus on the characterization of the synthetic CFOR.

As you asked, strain D05 is the engineered strain that co-express the free enzyme of CODH and FDH without the linker. Please refer to the Result section (page 6, line 126-page 7, line 128), Table S1, and Fig. S3. The formate productivity of strain D05 was 30-fold lower than the strain BCF13 as described in the manuscript (page 8, line 165-167). It read as follows:

“In the D05, formate production was detected under CO/CO₂ mix gas as a concentration of 0.1 ± 0.007 mmol/L at 60 min, which is 30-fold lower than the BCF13 (Fig. 2e).”

REVIEWERS' COMMENTS:

Reviewer #1 (Remarks to the Author):

I recommend accepting. All of my comments are fully addressed, I don't have additional comments.

Reviewer #3 (Remarks to the Author):

After careful reading the response letter and the revised manuscript, I think this revised manuscript neither addressed the questions nor meet publication standards. The author doesn't provide additional experimental evidence to convince the referee and reader.

1. The biggest highlight of the manuscript was that the magnetic property among [Fe-S] clusters contributed to the higher performance of the multi-enzyme systems with a linker. Unfortunately, the hypothesis proposed in the discussion section lacks experimental evidence in an earlier version. However, the authors agree that the suggested theory is not a proven hypothesis and removed the relevant content from both Introduction and Discussion in the revised manuscript. As they write in the discussion part "However, the principles of direct electron transfer by Fe-S fusion protein are unclear, and the validations remain further". Thus, I think the verification and interpretation of experimental phenomena are unclear and need more independent experiments. On the contrary, the author considers that "additional experiments are not currently being considered".

2. The author could not exclude the possibility that the great difference between multi-enzyme systems with linker and free enzyme may be caused by the rate of diffusion of the substrate and coenzyme, rather than the electron transfer.